# Associations of self-reported hearing problems with long-term trajectories of mental and functional health in middle-aged and older adults: The role of self-perceptions of aging

Markus Wettstein[1,2]*, Ann-Kristin Reinhard[2], Bettina Williger[3], Susanne Wurm[2]

1 Humboldt University Berlin, Berlin, Germany, 2 Department of Prevention Research and Social Medicine, Institute for Community Medicine, University Medicine, Greifswald, Germany, 3 Interdisciplinary Studies, University of Applied Science Landshut, Landshut, Germany

* markus.wettstein@med.uni-greifswald.de

## Abstract

Impaired hearing is a frequent stressful experience in later life. However, not all individuals affected by hearing problems exhibit restrictions in mental or functional health; there may be psychosocial resources that buffer the detrimental impact of impaired hearing to some extent. We investigate whether positive self-perceptions of aging (SPA, i. e. perceptions of more ongoing development and perceptions of less physical and social losses) do in fact constitute a compensatory resource and moderate between-person and within-person associations of self-reported hearing problems with trajectories of mental and functional health. Our sample comprised n = 9,705 participants in the German Ageing Survey (mean age = 62.41 years, SD = 11.58, range 40-93 years) who were assessed up to five times between 2008 and 2021. Based on longitudinal multilevel regression models, controlling for socio-demographic variables, hearing aid use, and chronic diseases, we found that individuals with overall more hearing problems reported poorer mental health. Moreover, with regard to the within-person associations, on measurement occasions when individuals reported more hearing problems, their mental and functional health was poorer. The association of more hearing problems with poorer functional health was weaker among individuals with lower perceptions of physical loss at baseline (i.e., year 2008). Among chronologically older adults, the association of more hearing problems with poorer functional health was weaker for those with higher perceptions of ongoing development at baseline. Our findings suggest that SPA related to fewer physical losses and to more ongoing development may buffer the negative impact of hearing problems on functional health.

**Data availability statement:** Given that this is a secondary data analysis, we are not in the position to share the data. The data from the German Ageing Survey can be obtained from the German Centre of Gerontology, Research Data Centre: https://www.dza.de/en/research/fdz/access-to-data. Our description of the measurement occasions that were used in the manuscript, data exclusions etc. will allow any researcher to build a dataset, based on the data received from the Research Data Centre of the German Centre of Gerontology, that will correspond to the dataset we used for analyses. We also provide the analytical code of our analyses to facilitate replication.

**Funding:** The project on which this publication is based was funded by the Innovation Committee of the Federal Joint Committee (G-BA) under the funding code 01VSF24037. The authors use data from the German Ageing Survey (DEAS), which was funded under Grant 301-6083-05/003*2 by the German Federal Ministry for Education, Family Affairs, Senior Citizens, Women, and Youth.

**Competing interests:** The authors have declared that no competing interests exist.

Impaired hearing is a highly stressful experience and a chronic condition that is particularly common in later life [1–4]. It affects many aspects of everyday life [5] and has a negative impact on mental and functional health [e.g., 6,7]. However, for some persons hearing problems are associated with severe health restrictions whereas others are less affected. In order to prevent negative consequences of hearing loss, it is crucial to identify important individual resources to mitigate its detrimental impact on health. Based on previous studies that pointed to the role of self-perceptions of aging (SPA) for mental and functional health [8], the present study examines the role of SPA in the association of self-reported hearing and trajectories of mental and functional health over up to 13 years in a large sample of middle-aged and older adults.

### Prevalence and consequences of hearing impairment

Worldwide about 65% of people aged 60 years and older are affected by at least mild hearing loss (≥ 25 dB hearing level [HL] across the frequencies 0.5, 1, 2, and 4 kHz), and 25% are affected by moderate to severe hearing loss (>=40 dB HL) [9]. Additionally, the risk of hearing loss increases with advancing age. According to the World Report on Hearing [5], about 11% of Europeans aged 60–69 years are affected by moderate to severe hearing loss; more than 40% of individuals in Europe aged 80 years and older reveal at least moderate hearing loss.

In the present study we focus on self-reported hearing impairment for the following reasons: although self-reported hearing impairment does not reveal a perfect congruency with objective hearing as measured by pure-tone audiometry, a relatively high concordance of objective vs. subjective hearing has been reported [e.g., 10,11–13]. Moreover, objective hearing tests may not accurately reflect everyday hearing problems and listening situations [14], which occur outside of the laboratory and reflect more than hearing acuity (e.g., speech comprehension in group meetings, when there is surrounding noise). Previous research has found that subjective hearing problems are more closely related to outcomes of mental health than objective hearing acuity [15–17]. Whereas older adults are less likely to report hearing complaints even when they are affected by objective hearing loss, the prevalence of subjective hearing complaints in midlife seems to be higher than the prevalence of objective hearing loss [1]. Associations of subjective hearing impairment with functional and mental health could therefore be different for middle-aged vs. older adults, as we will discuss in some more detail below.

Subjective hearing problems not only complicate and challenge social interactions and social activities [18], they are also associated with a greater risk of impaired functional health and frailty [6,19–22]. Moreover, impaired hearing is associated with a greater risk of cognitive impairment [23], lower well-being and poorer mental health [e.g., 24] such as greater depressive symptoms [7,25–28].

Impaired hearing thus seems to have negative consequences for mental and functional health. However, these consequences are not the same for everyone who is affected by hearing impairment. In the following section, we discuss potential moderating factors which either increase or reduce the association of impaired hearing with indicators of mental and functional health.

## Factors moderating the impact of impaired hearing on mental and functional health

It is crucial to identify individual resources that mitigate the detrimental impact of impaired hearing on health. In previous research, social support was found to be such a resource. Specifically, in a longitudinal study of 6,075 participants, West [29] found that greater social support was associated with reduced depressive symptoms among those who reported poorer hearing.

Another resource are intact cognitive abilities. Wettstein, Wahl [30] observed stronger cross-sectional associations between cognitive abilities and well-being in older adults with impaired hearing compared to older adults without sensory impairment. Investing in cognitive resources when sensory impairment sets in might thus contribute to maintaining greater well-being and better mental health. In addition, Heyl and Wahl [31] found that cognitive abilities are more strongly related to everyday functioning in older individuals with impaired hearing or vision compared to sensory unimpaired older individuals. Older adults with impaired hearing might thus be able to prevent restrictions in everyday functioning as long as they have sufficient cognitive resources available that may to some extent compensate for the impaired hearing.

Finally, coping strategies may play a compensatory role. In a qualitative study, Lash and Helme [32] identified a positive and accepting attitude towards one's own hearing problems as a strategy for maintaining psychological well-being and as a way of coping with negative consequences (such as depression) of hearing loss related stigma. In another study, higher scores on flexible goal adjustment as an indicator of coping were found to predict higher positive affect four years later in older adults with impaired hearing, but not in individuals with intact sensory functioning [33]. The coping strategy of flexible goal adjustment might thus become particularly adaptive for the maintenance of well-being and mental health after the onset of hearing loss.

To conclude, the fact that not all individuals whose hearing is impaired experience declines in mental and physical health seems to be due to interindividual differences in various factors that may act as compensatory resources, including social support and interaction, cognitive abilities, as well as coping strategies. However, most of the summarized findings were either based on cross-sectional data, or based on longitudinal data that comprised observation periods of less than five years. Thus, it remains unclear whether compensatory resources may reduce the detrimental *long-term* consequences of impaired hearing for outcomes of mental and functional health.

SPA are related to all of the moderating factors summarized above as well as to aspects of mental and functional health [34–38] and it could be that SPA operate as moderators on associations between hearing impairment and mental and functional health via coping strategies, social support, and other factors. The specific pathway might vary according to the specific SPA domain considered; for instance, self-efficacy, positive affect and optimism are most closely related to SPA of ongoing development, whereas aspects of social support, loneliness, and negative affect are particularly closely associated with SPA of social loss [39–41]. However, to our knowledge, the moderating role of SPA on associations of hearing impairment with functional and mental health has not been investigated so far; therefore, the aim of the present study is to investigate such SPA moderation effects.

## The relevance of self-perceptions of aging for health outcomes

SPA describe how individuals perceive and evaluate their own aging in different life domains [42–45] including different gains and losses [46]. More favorable SPA are associated with various beneficial developmental outcomes, such as higher well-being, better mental and physical health or lower mortality hazards [8]. These associations of more favorable SPA with better health outcomes are due to various behavioral, psychological and physiological mechanisms [47].

Positive SPA may not only be protective factors with regard to developmental outcomes such as mental or functional health, they might also be relevant as compensatory psychosocial moderators which counteract the detrimental impact of certain health risk factors such as hearing impairment. Using longitudinal data from the Health and Retirement Study, a study by Han [48] showed that the number of chronic illnesses in 2008 predicted more severe depressive symptoms two years later; this association was significantly stronger among participants who reported more negative SPA at baseline.

Based on another longitudinal study, Wurm, Warner [49] and Wolff, Schüz [50] investigated the moderating effect of SPA related to physical losses in the context of health events. SPA were found to moderate the relationship between health events and self-regulatory strategies to live a healthy lifestyle. Specifically, individuals with more negative SPA (related to physical losses) made less use of these strategies than those with less negative SPA [49]. Findings of Wolff, Schüz [50] add that individuals with negative SPA (related to physical losses) who experienced a serious health event showed more functional limitations 2.5 years later compared to individuals with less negative SPA who experienced a serious health event.

Based on a daily diary study, Bellingtier and Neupert [51] found that older adults with more positive SPA exhibited a lower reactivity to daily stressors than individuals with more negative SPA. These findings were supported by Witzel, Turner [52] in a microlongitudinal study across 100 days indicating that both the between- and the within-person association of greater perceived stress with more physical health symptoms was lower among individuals with more positive SPA. Kornadt, Albert [53] found that perceived ageism was related to lower life satisfaction in older adults, and this association was heightened among those with more negative SPA related to social losses. Finally, Levy, Slade [54] found that more positive SPA at baseline were associated with a lower risk of developing dementia over an observation period of four years, controlling for a broad set of relevant covariates such as depressive symptoms, cardiovascular diseases or diabetes. This effect was particularly strong among APOE ε4 carriers, who generally have a higher risk of developing dementia.

In conclusion, positive SPA may buffer the negative impact of certain risk factors on mental and physical health. However, the role of positive SPA as a multidimensional construct comprising both high gain-oriented perceptions and low loss-oriented perceptions as a health-protective resource for those who are affected by self-reported hearing impairment has so far not been investigated.

### Self-perceptions of aging as moderator of associations between hearing problems and health

There are several mechanisms that might explain the potentially moderating effect of SPA on the associations of self-reported hearing problems with mental and functional health. Specifically, experiencing hearing problems is highly stressful and positive SPA have been found to buffer the association of greater stress with affective well-being [51] and cardiovascular stress response [55]. Moreover, positive SPA, including those related to ongoing development, are related to preventive health behaviors such as physical activity [56–59], which is another "stress buffer" [60] and contributes to maintaining mental and functional health. Individuals with more favorable SPA might also be more likely to seek help (e.g., by getting hearing aids) or to use different communication strategies and thereby avoid negative consequences for mental and functional health. Conversely, individuals with more negative SPA might, as a consequence of age-related self-stigmatization [61,62], attribute the experience of hearing loss to their own aging and might believe that their hearing impairment is irreversible and untreatable. Finally, positive SPA are associated with various psychosocial resources such as control beliefs, self-efficacy, self-regulatory strategies and will to live [34–36,38,49,63]. These resources might additionally help to adjust to impaired hearing and to prevent or alleviate negative consequences for mental and functional health.

### The role of chronological age

The association of impaired hearing with functional and mental health, as well as the role of SPA, both as predictor and moderator, might also vary by chronological age. Among middle-aged adults, self-reported hearing loss is more frequent than audiometrically assessed hearing loss, whereas among older adults, the proportion of individuals affected by audiometrically assessed hearing loss is greater than the proportion of individuals with self-reported hearing loss [1]. It is possible that older adults consider impaired hearing as "normative" and might therefore perceive their hearing, in comparison to age peers, as intact, even when they are affected by a hearing impairment. In contrast, middle-aged adults might become aware of subtle changes in hearing which they interpret as "hearing loss", even though these subtle changes do not correspond to an objective hearing loss as assessed by audiometry.

From a theoretical perspective, one could assume that hearing loss at older ages is particularly detrimental for mental health, as socioemotional selectivity theory [64] states that with increasing age, close social relationships become more important, but hearing loss might challenge social interactions with emotionally close persons. However, empirical studies have shown that hearing impairment has a greater impact on the mental health of younger adults compared to older adults [49]. This could be due to an overall increased risk of hearing problems with advancing age and that deterioration in hearing among older individuals occurs steadily, whereas hearing loss in younger age groups is often linked to a sudden onset. Additionally, older adults may interpret hearing loss as a normal part of aging whereas the onset of impaired hearing in midlife might be regarded as an "off-time" experience [65] and challenge an individual's mental health.

The role of chronological age might be different in the context of functional health: As functional health grows more vulnerable with advancing age, each additional risk factor – such as the onset of hearing loss – might have a detrimental impact. In contrast, middle-aged adults have a lower risk of functional health decline and might be better able to compensate for the impact of risk factors such as hearing loss by means of higher biological plasticity and other resources. In the context of vision, for example, Wettstein, Spuling [66] found stronger associations between poorer self-rated vision and functional health among older compared to middle-aged adults.

As SPA become more self-relevant when individuals grow older and "increase in salience with older chronological age" [43], their impact on functional and mental health, but also their role as resources buffering the negative impact of impaired hearing on functional and mental health, might grow stronger with advancing age. Several studies indeed found that SPA are more strongly associated with health among chronologically older adults [67–70].

## The present study

In this study, we investigate the association of self-reported hearing problems and SPA with trajectories of mental and functional health over up to 13 years in a sample of middle-aged and older adults. Moreover, we analyze whether the association of self-reported hearing problems and health are moderated by SPA. One established indicator that takes into account the multidimensionality of SPA [40,43,71] as well as the conceptualization of lifespan development as a process comprising both gains and losses are the so-called aging-related cognitions scales (AgeCog scales) [41,72]. These scales consist of three subscales that comprise both losses (perceptions of social and physical losses) and gains (perceptions of ongoing development). To address the multidimensionality of SPA, we will focus on all three subscales of the AgeCog scales. As with more favorable SPA in general, greater perceptions of ongoing development and lower perceptions of social and physical losses are related to better mental and physical health [39–41,72–74].

As the subjective experience of hearing problems is highly dynamic and subject to change across time, we will consider hearing problems as a time-varying predictor and distinguish – following common practice [75–77] – between-person associations and within-person associations of hearing problems with trajectories of mental and functional health. According to Hamaker [76], "there are scenarios in which the interest is primarily in understanding the process as it unfolds within an individual over time, whereas there are also study areas where the main interest is in understanding why individuals are different from each other in lasting ways" (p. 5). In this study, the focus will be on both scenarios. Significant between-person associations indicate to what extent individuals with overall more hearing problems are at risk for impaired mental or functional health, whereas significant within-person associations indicate to what extent the onset or a deterioration of hearing problems is associated with a decrease in mental and functional health. This differentiation is important, as these between- vs. within-person associations have different implications regarding interventions. Specifically, a significant within-person association of more hearing problems with poorer functional and mental health indicates that counteracting the within-person increase in hearing problems may be particularly important to promote functional and mental health, whereas a significant between-person association rather implies that restoring hearing among those with overall more hearing problems might help to maintain or improve functional and mental health.

Our hypotheses are as follows:

1. Between-person effect of hearing problems: We assume that individuals with overall more hearing problems report poorer mental and functional health and exhibit a steeper decline in mental and functional health over time.

2. Within-person effect of hearing problems: We assume that individuals report poorer functional and mental health on measurement occasions when they report more problems with hearing.

3. Effect of SPA: We expect that individuals with more positive SPA related to ongoing development and less negative SPA related to social and physical losses report better mental and functional health and exhibit a less steep decline in mental and functional health over time.

4. SPA moderation effects: We assume that the between-person and within-person associations of more hearing problems with poorer mental and functional health (as described in H1 and H2) are weaker among those individuals with higher scores on perceptions of ongoing development and stronger among those with higher scores on social and physical losses.

5. The role of chronological age: We assume that more hearing problems are less strongly associated with poorer mental health in chronologically older adults, but that the association of more hearing problems with poorer functional health are stronger among chronologically older individuals. Moreover, we assume the associations between SPA and mental and functional health are stronger among chronologically older individuals. Finally, we expect chronological age to moderate the SPA moderation effects, with stronger SPA moderation effects among chronologically older adults.

6. Given the multidimensionality of SPA, it is possible that the associations of SPA with functional and mental health as well as the moderating effects of the SPA vary according to which SPA domain is considered. For instance, the "stereotype-matching effect" [78] postulates that age stereotypes that refer to a specific outcome are more closely related to this outcome than more "distal" age stereotypes. Applying this matching effect to the context of SPA, it could be assumed that SPA of physical losses are more closely related to functional health than the other SPA and they might also be stronger moderators of associations between hearing problems and functional health. However, prior evidence, particularly regarding SPA-domain-specific moderation effects, is restricted. Therefore, we will investigate this research question of potential domain specific effects in an exploratory manner.

## Materials and methods

The analytic code for the following analyses is documented in online supplemental material. The design, hypotheses, and analytic plan of the current manuscript were not preregistered. In the following section, we state how we determined the sample size, and we report any data exclusions, all data preparatory steps, and all measures that were used for our analyses.

### Sample

Data from the German Ageing Survey (Deutscher Alterssurvey) [79–82] were used. The German Ageing Survey is a cohort-sequential study of a nationally representative sample of middle-aged and older adults (i. e., aged 40 years and older at the time of their first assessment) residing in Germany. The first sample was assessed in 1996, with follow-up assessments in 2002, 2008, 2011, 2014, 2017, 2020 and 2020/21. Additional samples were drawn in 2002, 2008, and 2014 and re-assessed on the later measurement occasions.

Ethical approval was not obtained because this is not mandatory for general surveys in Germany. The German Ageing Survey does not use any invasive methods. The survey maintains an academic advisory board which ensures its scientific quality. Prior to study participation, all study participants provide verbal informed consent which is documented by the interviewer. In addition, study participants also provide written informed consent that they will be re-contacted for future survey participations (panel consent).

Study inclusion criteria are that individuals have to reside in Germany and that they have to be able to understand and speak German.

For the current study, we used data from 2008 to 2020/2021 (2008, 2011, 2014, 2017, winter 2020/2021), with 2008 representing the baseline. We did not include measurement occasions prior to 2008, as the assessment of hearing problems was changed from 2002 to 2008, but then remained consistent from 2008 on. Several variables that are relevant for our study were not assessed in summer 2020 which was an additional measurement occasion with a focus on experiences during the COVID-19 pandemic. Therefore, we did not include this measurement occasion, but instead we included the subsequent measurement occasion which took place in winter 2020/2021. All individuals who provided valid scores on the study variables (hearing problems, mental and functional health, covariates) on at least one measurement occasion between 2008 and 2020/201 were included in the sample ($n = 9{,}705$). In our study sample, 5.8% ($n = 567$) of the participants reported migration background, and 98.2% ($n = 9{,}528$) had a German citizenship.

## Measures

**Mental health.** We used depressive symptoms as an indicator for mental health. Depressive symptoms were assessed using the 15-item German version of the CES-D scale [83,84]. Participants were asked to rate the frequency of depressive symptoms experienced during the past week on a scale from 0 – *Rarely or none of the time* to 3 – *Most or all of the time.* These depressive symptoms comprise aspects of anhedonia/lack of positive affect (e.g., "During the last week I felt depressed"), interpersonal aspects (e.g., "During the last week I felt lonely"), and somatic complaints (e.g., "During the last week my sleep was restless"). Items were aggregated into a sum score, with higher scores indicating more severe depressive symptoms (Cronbach's α 2008–2021: .86, .86, .85, .86, .84).

**Functional health.** Functional health was assessed with the 10-item subscale physical functioning of the SF-36 [85,86]. Participants evaluated their current impairment with regard to different activities of daily living (ADLs) (e.g., bending, kneeling, stooping) on a scale from 1 – *Yes, limited a lot* to 3 – *No, not limited at all*. Items were transformed and combined into a sum score according to the SF-36 coding guidelines [85,86], with a resulting 0–100 score range. Higher ratings indicate better functional health.

**Hearing problems.** Problems with hearing were assessed with two items, referring to hearing on the telephone ("Do you have hearing problems with phone calls [even when using a hearing aid]?") and in groups ("Do you have problems with hearing in group meetings with four or more people [even when using a hearing aid])?"). Similarly, worded items have been used in previous studies [e.g., 87–89]. Limitations were rated on a 4-point Likert scale (1 – *No difficulties* to 4 – *Impossible*). As both items were strongly interrelated across the measurement occasions ($.51 \le r \le .61$), we used the mean score of both items as an indicator of hearing problems.

**Self-perceptions of aging (SPA).** SPA at baseline (2008) were measured with the Aging-Related Cognitions Questionnaire (AgeCog) [41,72]. The AgeCog Scales comprise one subscale representing gain-related SPA (*ongoing development)* and two subscales representing loss-related SPA (*physical and social loss*). Items of all subscales included the stem "Aging means to me…" followed by a domain-specific statement such as "…that I continue to make plans" (*ongoing development*), "…that I am less energetic and fit" (*physical losses*) or "…that I feel less needed" (*social loss*). Each subscale contained 4 items, which were rated on a 4-point Likert scale (1 = *Strongly agree* to 4 = *Strongly disagree*). For further analyses a mean score was computed using recoded items, with higher ratings indicating higher values on each subscale (Cronbach's α ongoing development 2008 and 2014: .82 and .79; Cronbach's α social loss 2008 and 2014: .75 and .72; Cronbach's α physical loss 2008 and 2014: .77 and .77).

**Covariates.** In our analyses, we controlled for factors that are associated with both hearing problems and mental/functional health and which might therefore be relevant confounders. Specifically, sociodemographic indicators such as age, sex, education, or region of residence (West or East Germany) are associated with hearing problems [90–92] as

well as with SPA [39–41,73]. Education was measured based on the International Standard Classification of Education (ISCED) [93] coding; this coding integrates school and professional education into one indicator that differentiates between four educational levels (low, medium, elevated, and high education). Similarly, we controlled for the number of self-reported diseases because these are related to both hearing problems [90–92] and SPA [39–41,73]. We also included hearing aid use as a covariate; hearing aids can improve hearing problems, but they might also be regarded as an "aging body reminder" [94] and affect individuals' SPA. Finally, year of study entry (1996, 2002, 2008 or 2014) was controlled for because individuals who entered the survey already in 1996 and still took part in 2008 might be more positively selected than individuals who first took part in 2008; moreover, individuals who entered the survey in different years also belong to different birthyear cohorts, so that this covariate can also be regarded as an indicator of cohort effects. All covariates were specified as time-invariant.

## Statistical analyses

We computed longitudinal multilevel regression models [95,96] to investigate the associations of hearing problems with trajectories of mental and functional health as well as the moderating role of gain- and loss-related SPA. The time metrics for trajectories of mental and functional health was "time in study" (in years). We specified separate models for each outcome, which were adjusted for age, sex, education, hearing aid use, number of self-reported diseases, region of residence, and year of study entry.

Hearing problems were specified as time-varying predictors. Following common practice [77], the between-person component of hearing problems (bphearing) corresponds to each individual's mean across their available measurement occasions, whereas the within-person component of hearing problems (wphearing) represents measurement occasion-specific deviations of an individual from their own mean. The model equation for functional health was as follows (and the equation for mental health as outcome was the same):

$\text{Functionalhealth}_{it} = \gamma_{00} + \gamma_{01}(\text{Age}_i) + \gamma_{11}(\text{wphearing}_{it}) + \gamma_{02}(\text{bphearing}_i) + \gamma_{12}(\text{Age}_i * \text{wphearing}_{it}) + \gamma_{03}(\text{Age}_i * \text{bphearing}_i) + \gamma_{04}(\text{SPA social loss}_i) + \gamma_{13}(\text{SPA social loss}_i * \text{wphearing}_{it}) + \gamma_{05}(\text{SPA social loss}_i * \text{bphearing}_i) + \gamma_{14}(\text{SPA social loss}_i * \text{age} * \text{wphearing}_{it}) + \gamma_{06}(\text{SPA social loss}_i * \text{age} * \text{bphearing}_i) + \gamma_{07}(\text{SPA physical loss}_i) + \gamma_{15}(\text{SPA physical loss}_i * \text{wphearing}_{it}) + \gamma_{08}(\text{SPA physical loss}_i * \text{bphearing}_i) + \gamma_{16}(\text{SPA physical loss}_i * \text{age} * \text{wphearing}_{it}) + \gamma_{09}(\text{SPA physical loss}_i * \text{age} * \text{bphearing}_i) + \gamma_{08}(\text{SPA ongoing development}_i) + \gamma_{17}(\text{SPA ongoing development}_i * \text{wphearing}_{it}) + \gamma_{09}(\text{SPA ongoing development}_i * \text{bphearing}_i) + \gamma_{18}(\text{SPA ongoing development}_i * \text{age} * \text{wphearing}_{it}) + \gamma_{010}(\text{SPA ongoing development}_i * \text{age} * \text{bphearing}_i) + \gamma_{...}(\text{covariates}_i) + \gamma_{19}(\text{time}_{it}) + \gamma_{100}(\text{Age}_i * \text{time}_{it}) + \gamma_{101}(\text{bphearing}_i * \text{time}_{it}) + \gamma_{102}(\text{Age}_i * \text{bphearing}_i * \text{time}_{it}) + \gamma_{103}(\text{SPA social loss}_i * \text{time}_{it}) + \gamma_{104}(\text{SPA social loss}_i * \text{bphearing}_i * \text{time}_{it}) + \gamma_{105}(\text{SPA social loss}_i * \text{age} * \text{bphearing}_i * \text{time}_{it}) + \gamma_{106}(\text{SPA physical loss}_i * \text{time}_{it}) + \gamma_{107}(\text{SPA physical loss}_i * \text{bphearing}_i * \text{time}_{it}) + \gamma_{108}(\text{SPA physical loss}_i * \text{age} * \text{bphearing}_i * \text{time}_{it}) + \gamma_{109}(\text{SPA ongoing development}_i * \text{time}_{it}) + \gamma_{110}(\text{SPA ongoing development}_i * \text{bphearing}_i * \text{time}_{it}) + \gamma_{111}(\text{SPA ongoing development}_i * \text{age} * \text{bphearing}_i * \text{time}_{it}) + \gamma_{...}(\text{covariates}_i * \text{time}_{it}) + u_{0i} + u_{1i}\text{time}_{it} + r_{it}$

For the sake of model parsimony, only those interactions including chronological age were maintained in the model that were statistically significant at $p < .01$, whereas all nonsignificant interaction terms with chronological age were trimmed from the final models. Moreover, we did not specify interactions of within-person hearing with time in study, as there is no straightforward interpretation for such interaction terms. We chose the significance threshold of $p < .01$ for all effects, given our large sample size and for the sake of avoiding an over-interpretation of negligible effects.

Sampling weights were not used; instead, Full Information Maximum Likelihood Estimation was used to handle potentially selective dropout, and dropout-informative covariates including chronic diseases, were included in the models.

**Statistical power.** No a-priori power analyses were conducted. As a sample size of $n$ close to 10,000 with almost 25,000 observations was available, statistical power should be sufficient to detect small to medium effect sizes.

## Results

### Sample description

Sample characteristics of the current study can be found in Table 1. The mean age in the sample was 62.41 years. About 50% in the study sample were women. Depressive symptoms were higher among individuals reporting more hearing problems ($r=.16$), lower SPA of ongoing development ($r=-.30$), and higher SPA of social loss ($r=.34$) as well as physical loss ($r=.29$). Similarly, functional health was poorer among those reporting more hearing problems ($r=-.21$) and those with lower scores on SPA of ongoing development ($r=.33$) as well as higher scores on SPA of social loss ($r=-.21$) and physical loss ($r=-.39$). The associations of more hearing problems with lower SPA of ongoing development ($r=-.13$), higher SPA of social loss ($r=.10$) as well as of physical loss ($r=.13$) were rather small, whereas the SPA scales were moderately interrelated (all $r > |.40|$).

### Associations of hearing problems and self-perceptions of aging with trajectories of mental health

Findings from the longitudinal multilevel regression models are shown in Table 2. Mean-level change in depressive symptoms was not significant ($\beta_{time}=0.005$, SD$=0.028$, $p=.87$), but the random slope effect indicated significant interindividual variation in the trajectories of depressive symptoms. As to the effects of hearing problems, the significant between-person effect of hearing problems indicated that individuals with overall more hearing problems scored higher on depressive symptoms ($\beta_{bphearing}=1.39$, SD$=0.19$, $p<.0001$; see black line in Fig 1A). On the within-person level, individuals reported more depressive symptoms on measurement occasions where they reported more hearing problems ($\beta_{wphearing}=0.90$, SD$=0.17$, $p<.0001$; see red line in Fig 1A). Individuals with different levels of overall hearing problems did not significantly differ regarding the extent of within-person change in depressive symptoms ($\beta_{bphearing*time}=-0.02$, SD$=0.04$, $p=.68$).

**Table 1. Descriptive statistics at baseline assessment (2008) and intercorrelations for study measures.**

| | 1 | 2 | 3 | 4 | 5 | 6 | 7 | 8 | 9 | 10 | 11 | 12 | 13 | 14 | 15 |
|---|---|---|---|---|---|---|---|---|---|---|---|---|---|---|---|
| (1) Depressive Symptoms (0–42) | 1 | | | | | | | | | | | | | | |
| (2) Functional Health (0–100) | −.41 | 1 | | | | | | | | | | | | | |
| (3) Hearing Problems (1–4) | .16 | −.21 | 1 | | | | | | | | | | | | |
| (4) SPA Ongoing Development (1–4) | −.30 | .33 | −.13 | 1 | | | | | | | | | | | |
| (5) SPA Social Losses (1–4) | .34 | −.21 | .10 | −.42 | 1 | | | | | | | | | | |
| (6) SPA Physical Losses (1–4) | .29 | −.39 | .13 | −.42 | .41 | 1 | | | | | | | | | |
| (7) Age (40–93) | .02 | −.33 | .21 | −.25 | .05 | .18 | 1 | | | | | | | | |
| (8) % Women | .10 | −.07 | −.06 | .02 | −.01 | −.03 | −.09 | 1 | | | | | | | |
| (9) Education (1–4) | −.12 | .19 | −.04 | .18 | −.12 | −.09 | −.11 | −.18 | 1 | | | | | | |
| (10) Study Entry: % 2002 | −.04 | −.00 | −.01 | −.00 | −.01 | −.01 | .06 | −.00 | .01 | 1 | | | | | |
| (11) Study Entry: % 2008 | −.02 | .05 | −.05 | −.02 | .03 | .03 | −.06 | −.01 | −.06 | −.26 | 1 | | | | |
| (12) Study Entry: % 2014 | .06 | −.04 | .06 | .03 | −.02 | −.03 | −.04 | .01 | .07 | −.24 | −.72 | 1 | | | |
| (13) % East Germany | .05 | −.05 | −.02 | −.08 | −.01 | .06 | .00 | .02 | .07 | .00 | .04 | −.04 | 1 | | |
| (14) % Hearing Aid | .05 | −.14 | .38 | −.07 | .03 | .08 | .23 | −.05 | −.04 | −.00 | −.03 | .02 | −.01 | 1 | |
| (15) Chronic Diseases (0–11) | .29 | −.45 | .33 | −.25 | .21 | .33 | .37 | −.03 | −.10 | −.01 | −.06 | .05 | .02 | .24 | 1 |
| *M* or % | 6.43 | 83.62 | 1.22 | 2.90 | 1.85 | 2.79 | 62.41 | 49.47% | 2.56 | 8.16% | 43.85% | 39.85% | 34.19% | 6.97% | 2.38 |
| *SD* | 6.01 | 22.25 | 0.44 | 0.59 | 0.57 | 0.55 | 11.58 | | 0.96 | | | | | | 1.86 |

SPA$=$Self-perceptions of aging. *M*$=$mean, *SD*$=$standard deviation. % scores indicate the percentage of women, of individuals who entered the survey in 2002, 2008, 2014, and of individuals using hearing aids, respectively. Education was coded as 1$=$low, 2$=$medium, 3$=$elevated, and 4$=$high.

Intercorrelations of $|r|=.03$ or above differ statistically significantly from zero at $p<.01$.

**Table 2. Growth models of mental health (depressive symptoms) and functional health.**

| | Depressive Symptoms | | Functional Health | |
|---|---|---|---|---|
| | Estimate | SE | Estimate | SE |
| **Fixed effects** | | | | |
| Intercept | 6.400*** | 0.194 | 86.667*** | 0.691 |
| Age | −0.066*** | 0.005 | −0.261*** | 0.018 |
| Women | 1.104*** | 0.104 | −3.324*** | 0.368 |
| Hearing Aid | −0.494 | 0.224 | 0.242 | 0.789 |
| Chronic Diseases | 0.638*** | 0.032 | −3.207*** | 0.115 |
| Education | −0.322*** | 0.055 | 2.336*** | 0.196 |
| East Germany | 0.383*** | 0.108 | −1.109** | 0.383 |
| SPA Ongoing Development | −1.309*** | 0.108 | 3.403*** | 0.378 |
| SPA Social Losses | 2.002*** | 0.104 | −0.348 | 0.369 |
| SPA Physical Losses | 1.104*** | 0.111 | −8.583*** | 0.391 |
| Hearing Problems (bp) | 1.387*** | 0.189 | −1.134 | 0.666 |
| Hearing Problems (wp) | 0.899*** | 0.169 | −2.756*** | 0.481 |
| Age*SPA Ongoing Development | −0.023** | 0.007 | 0.181*** | 0.0291 |
| Age*SPA Physical Loss | − | − | −0.300*** | 0.031 |
| Study Entry 2002 | 0.084 | 0.246 | −0.716 | 0.877 |
| Study Entry 2008 | 0.370 | 0.191 | 0.201 | 0.681 |
| Study Entry 2014 | 0.905*** | 0.194 | −1.988** | 0.689 |
| SPA Ongoing Development*Hearing Problems (bp) | −0.180 | 0.235 | −0.506 | 0.933 |
| SPA Social Loss*Hearing Problems (bp) | 0.568 | 0.233 | −0.2293 | 0.8214 |
| SPA Physical Loss*Hearing Problems (bp) | 0.005 | 0.256 | −2.742** | 0.922 |
| SPA Ongoing Development*Hearing Problems (wp) | 0.165 | 0.230 | −0.256 | 0.658 |
| SPA Social Loss*Hearing Problems (wp) | 0.225 | 0.229 | −0.439 | 0.658 |
| SPA Physical Loss*Hearing Problems (wp) | 0.294 | 0.239 | −1.00 | 0.685 |
| Age*SPA Ongoing Development* Hearing Problems (bp) | − | − | 0.184** | 0.062 |
| Age*SPA Ongoing Development* Hearing Problems (wp) | − | − | − | − |
| Time | 0.005 | 0.028 | −0.887*** | 0.096 |
| Age*Time | 0.008*** | 0.001 | −0.045*** | 0.004 |
| Women*Time | 0.025 | 0.017 | 0.007 | 0.059 |
| Hearing Problems (bp)*Time | −0.015 | 0.035 | −0.096 | 0.123 |
| Hearing Aid*Time | −0.061 | 0.041 | 0.015 | 0.139 |
| Chronic Diseases*Time | −0.001 | 0.006 | −0.044* | 0.019 |
| Education*Time | −0.006 | 0.009 | 0.095** | 0.031 |
| East Germany*Time | 0.012 | 0.018 | −0.014 | 0.061 |
| SPA Ongoing Development*Time | 0.009 | 0.018 | −0.078 | 0.062 |
| SPA Social Loss*Time | −0.136*** | 0.018 | 0.064 | 0.063 |
| SPA Physical Loss*Time | 0.020 | 0.018 | −0.043 | 0.062 |
| Age*SPA Social Loss*Time | − | − | 0.013** | 0.005 |
| SPA Ongoing Development*Hearing Problems (bp)*time | 0.057 | 0.044 | −0.298 | 0.152 |
| SPA Social Loss*Hearing Problems (bp)*time | −0.111 | 0.046 | −0.077 | 0.165 |
| SPA Physical Loss*Hearing Problems (bp)*time | 0.078 | 0.046 | 0.093 | 0.161 |
| Age*SPA Social Loss*Hearing Problems (bp)*time | | | 0.058*** | 0.010 |
| Study Entry 2002*time | 0.002 | 0.033 | −0.041 | 0.114 |

*(Continued)*

**Table 2.** (Continued)

| | Depressive Symptoms | | Functional Health | |
|---|---|---|---|---|
| | **Estimate** | **SE** | **Estimate** | **SE** |
| Study Entry 2008*time | −0.009 | 0.026 | −0.213 | 0.091 |
| Study Entry 2014*time | −0.077 | 0.031 | 0.328** | 0.102 |
| Variance Intercept | 10.611*** | 0.402 | 202.06*** | 4.782 |
| Variance Slope | 0.043*** | 0.008 | 1.362*** | 0.095 |
| Cov. Intercept, slope | −0.089 | 0.050 | 0.732 | 0.588 |
| **Variance explained** | 6.7% | | 25.7% | |

$N$ = 9,705 who provided 24,689 observations. SPA = Self-perceptions of aging, wp = within–person. bp = between–person. Cov. = covariance.

Unstandardized estimates and standard errors are presented. Age was grand–mean–centered at 62.4 years. **$p$ < .01. ***$p$ < .001.

With respect to the effects of SPA, individuals with lower scores on ongoing development ($\beta_{ongoingdevelopment}$ = −1.31, SD = 0.11, $p$ < .0001) and with higher scores on social loss ($\beta_{socialloss}$ = 2.00, SD = 0.10, $p$ < .0001) and physical loss ($\beta_{physicalloss}$ = 1.10, SD = 0.11, $p$ < .0001) reported more depressive symptoms. With regard to longitudinal effects, participants with higher social loss scores exhibited a less steep increase in depressive symptoms ($\beta_{socialloss*time}$ = −0.14, SD = 0.02, $p$ < .0001).

One interaction between age and SPA reached statistical significance. Specifically, the association of higher SPA related to ongoing development with lower depressive symptoms was stronger among chronologically older adults ($\beta_{age*ongoingdevelopment}$ = −0.02, SD = 0.01, $p$ < .001; see Fig 2).

As to the effects of the covariates, an older age, being male, fewer chronic conditions, higher levels of education and living in West Germany were associated with fewer depressive symptoms at baseline. Individuals who had entered the survey later, i.e., in 2014, had higher scores on depressive symptoms at baseline. Increase in depressive symptoms over time was steeper among chronologically older individuals.

## Associations of hearing problems and self-perceptions of aging with trajectories of functional health

Functional health revealed a significant mean-level decrease over time ($\beta_{time}$ = −0.89, SD = 0.10, $p$ < .0001; Table 2). On the between-person level, overall more hearing problems were not significantly associated with poorer functional health at baseline ($\beta_{bphearing}$ = −1.13, SD = 0.67, $p$ = .08). Moreover, they were not significantly associated with change in functional health ($\beta_{bphearing*time}$ = −0.10, SD = 0.12, $p$ = .43). With regard to the within-person association, on measurement occasions when individuals reported more hearing problems, they had poorer functional health scores ($\beta_{wphearing}$ = −2.76, SD = 0.48, $p$ < .0001; see Fig 1B).

Two of the three SPA dimensions were also significantly associated with functional health. Specifically, functional health at baseline was higher among those scoring higher on SPA of ongoing development ($\beta_{ongoingdevelopment}$ = 3.40, SD = 0.38, $p$ < .0001) and lower on SPA of physical loss ($\beta_{physicalloss}$ = −8.58, SD = 0.39, $p$ < .0001). Both associations were stronger among chronologically older adults ($\beta_{ongoingdevelopment*age}$ = 0.18, SD = 0.03, $p$ < .0001; $\beta_{physicalloss*age}$ = −0.30, SD = 0.03, $p$ < .0001; see Fig 3A and B). However, all three SPA domains were not significantly related with change in functional health over time.

Several interactions between hearing problems and SPA reached statistical significance: The association of overall more hearing problems with poorer functional health at baseline was buffered among those with higher scores on SPA of ongoing development, but only among chronologically older adults ($\beta_{ongoingdevelopment*age*bphearing}$ = 0.18, SD = 0.06, $p$ = .003). In contrast, the association of poorer overall hearing problems with poorer functional health was stronger among those with higher perceptions of physical losses ($\beta_{physicalloss*bphearing}$ = −2.74, SD = 0.92, $p$ = .003; Fig 4). Finally, among chronologically

A

B

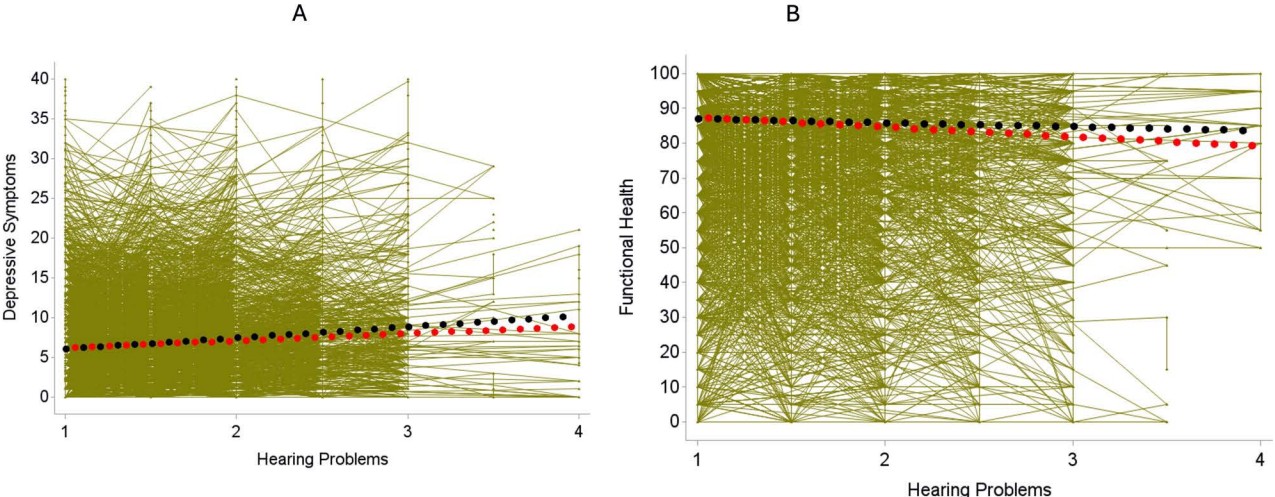

**Fig 1. Illustrating the between-person association (black line) and within-person association (red line) of hearing problems with depressive symptoms and functional health.** The green lines illustrate individual associations of all study participants. Single green points which are not connected to other points represent individuals who only took part once in the survey and thus provided no longitudinal information The x-axis shows the extent of hearing problems, ranging from 1 (no problems) to 4 (hearing impossible). Left figure (A): Individuals with overall poorer hearing report more depressive symptoms (black line), and on measurement occasions when individuals report more hearing problems, they exhibit more depressive symptoms (red line). Right figure (B): Illustrating the between-person association (black line) and within-person association (red line) of hearing problems with functional health. Individuals with overall poorer self-reported hearing do not differ from individuals with better hearing regarding their functional health (black line), but on measurement occasions when individuals report more hearing problems, they also report poorer functional health (red line).

older adults, higher perceptions of social loss were associated with a less steep decline in functional health over time ($\beta_{age*socialloss*time} = 0.01$, SD = 0.01, $p = .008$); this association was stronger among those with poorer overall hearing as compared to those with better overall hearing ($\beta_{age*socialloss*bphearing*time} = 0.06$, SD = 0.01, $p < .001$).

Regarding the effects of the covariates, baseline functional health was significantly poorer among older adults, women, individuals with more chronic diseases, with lower levels of education, and among persons living in East Germany. Functional health was also lower among those who had entered the survey in 2014 compared to those who had entered the survey before 2014. Decline in functional health was steeper among chronologically older persons, as well as in those individuals with more chronic diseases and lower levels of education. Decline was less steep in those whose first study participation had been in 2014.

## Discussion

In this study, we investigated the within- and between-person associations of hearing problems with trajectories of mental and physical health, as well as the moderating role of self-perceptions of aging, based on a study sample of middle-aged and older adults who were assessed up to five times over 13 years. In the following, we will summarize and discuss our major findings, their potential implications, as well as the limitations of the present study.

### Associations of hearing problems with mental and functional health

Controlling for various socio-demographic factors and for chronic diseases, we found a robust association between hearing problems and both mental and functional health. On the between-person level, individuals with overall more hearing problems reported more depressive symptoms. On the within-person level, on measurement occasions when individuals reported more hearing problems than their average, they also revealed elevated depressive symptoms and poorer

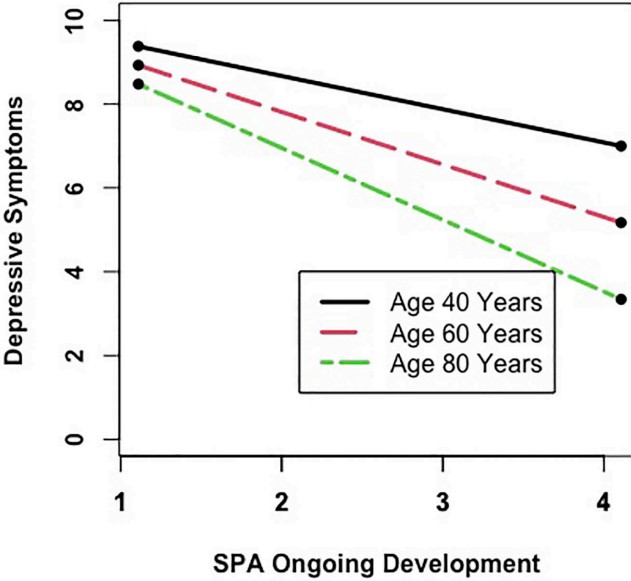

**Fig 2. The moderating effect of age on the association between SPA related to ongoing development and depressive symptoms.** Chronologically older individuals report lower levels of depressive symptoms than chronologically younger individuals. Higher scores on SPA of ongoing development are related to lower depressive symptoms, and this association is stronger among chronologically older individuals. The range of the y-scale of this figure was restricted to 0-10 for a better illustration of the age moderation effect. SPA = Self-perceptions of aging.

functional health. These associations of hearing problems with functional [6,19–22] and mental health [7,25–28] are in line with previous research and supported our first and second hypotheses; the only exception was the non-significant between-person hearing effect on functional health, which might have failed to reach statistical significance due to the covariates we controlled for. The within-person associations describe associations between concurrent changes, so firm causal conclusions cannot be drawn. For instance, rather than acting as a risk factor for poorer mental and functional health, hearing problems could result from poorer mental and physical health; specifically, elevated depressive symptoms might lead to a tendency of negative self-evaluations, potentially including one's own hearing, and as part of a "halo effect", individuals with functional health restrictions might also rate their hearing less favorably. Indeed, previous research has identified anxiety and depressive symptoms as psychosocial risk factors for hearing problems [97–99]. It could also be that there are mutual associations between hearing problems, mental health and functional health, so that preventing hearing loss, or alleviating hearing problems, e.g., by means of providing hearing aids, could contribute to maintaining high levels of mental and functional health (for the contribution of hearing aids to cognitive health, see [e.g., 23]), and promotion of mental and functional health could be beneficial for the maintenance of hearing health.

### Associations of self-perceptions of aging with mental and functional health

As expected, greater perceptions of ongoing development as well as lower perceptions of physical and social loss were related to better mental and functional health. However, regarding the health trajectories, only SPA of social loss was significantly related to change in depressive symptoms, with those scoring higher on SPA of social loss exhibiting less increase in depressive symptoms over time. This finding is counterintuitive, but it could be because individuals with higher SPA of social loss scores had already more depressive symptoms at baseline, so that their depressive symptoms did not increase any further over time. Alternatively, although we used Full Information Maximum Likelihood Estimation in order to also include the information provided by participants who dropped out of the study, individuals with higher SPA of social

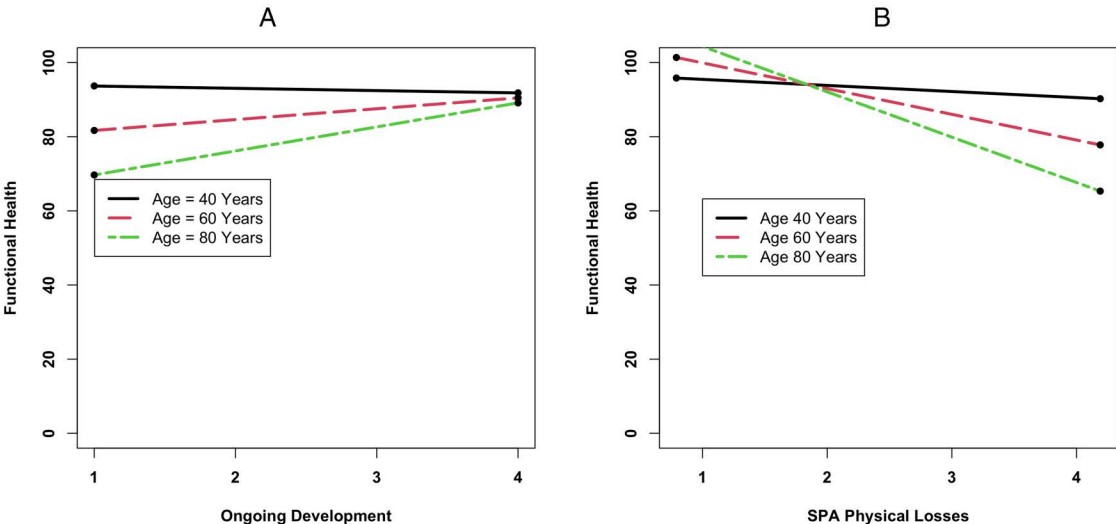

**Fig 3. The moderating effect of age on the association between SPA related to ongoing development and functional health and on the association between SPA related to physical loss and functional health.** Left figure (A): Chronologically older individuals report poorer functional health than chronologically younger individuals. Higher scores on SPA of ongoing development are related to better functional health, and this association is stronger among chronologically older individuals. Right figure (B): Chronologically older individuals report poorer functional health than chronologically younger individuals. Higher scores on SPA of physical loss are related to poorer functional health, particularly among chronologically older adults. SPA = Self-perceptions of aging.

loss scores and those with increasing depressive symptoms over time might have had a higher likelihood to drop out of the study.

The cross-sectional associations between SPA with hearing problems correspond to other studies which observed that more favorable SPA are related to better mental and functional health [8,100,101]. However, the associations we identified do not necessarily mean that SPA are antecedents of mental and functional health; health conditions, including depressive symptoms, were found to predict subsequent SPA in a recent study [102]. However, the pathway from SPA to health outcomes seems to be stronger than the reversed pathway [103,104]. Therefore, promoting positive SPA by means of interventions [see for instance 105–107], could improve middle-aged and older adults' mental and functional health.

With one exception noted above, we did not observe associations of SPA with changes in mental and functional health. However, we only included SPA assessed at baseline; it is possible that changes in SPA, rather than their levels, are associated with changes in mental and functional health.

### The moderating role of self-perceptions of aging

As there is empirical evidence suggesting that the detrimental impact of hearing problems on mental and functional health can be buffered by certain psychosocial resources [30,31,108–110], we investigated if SPA are in fact a resource that moderates associations between hearing problems and health trajectories. Indeed, overall more hearing problems were less strongly related to poorer functional health at baseline among individuals with lower perceptions of physical loss.

The role of SPA as "buffer" in the context of health risk factors – with positive SPA being associated with a reduced detrimental impact of risk factors on health outcomes – has been reported in previous research [51–54]. We extended these findings to the context of hearing problems. However, evidence for a buffering effect with regard to *changes* in mental and functional health as opposed to health *levels*, was limited, and thus provided only partial support for hypothesis 4. This might be again because we did not investigate changes in SPA and how they are associated with mental and functional health changes. Further research is thus needed to analyze the role of time-varying SPA as potential moderator in the

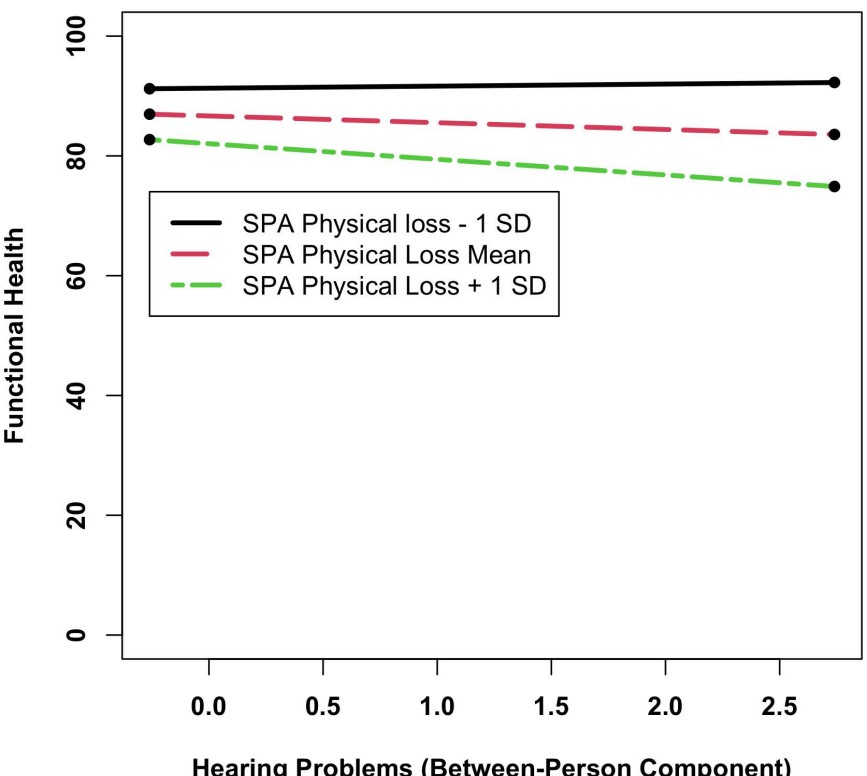

**Fig 4. The moderating effect of SPA related to physical loss on the association between hearing problems and functional health.** Higher scores of SPA of physical loss are associated with poorer functional health. Only among those with higher SPA of physical loss scores, more hearing problems are associated with poorer functional health. SPA = Self-perceptions of aging.

association between hearing problems and trajectories of mental and functional health. Moreover, we did not investigate mediators of the buffering effects of SPA – which could be, for instance, behavioral (e.g., individuals with positive SPA may seek out the best treatment against hearing loss) or psychological (e.g., persons with positive SPA may have higher psychological resources, such as self-efficacy, coping with hearing loss) – which require more investigation. Despite these open questions, positive SPA are, according to our findings, related to better mental and functional health. Additionally, they might be a promising resource against the detrimental impact of hearing problems on mental and functional health, which is – given the high prevalence of hearing loss in old age – a noteworthy finding and underlines the importance of efforts to promote positive views on aging both on a societal as well as on an individual level.

### The moderating role of age

Contrary to our expectation in hypothesis 5, we did not find that hearing problems are less strongly associated with depressive symptoms among chronologically older compared to chronologically younger adults. Also, the association of hearing problems with functional health did not vary according to chronological age. Age-differential effects might have been found if the study sample had also comprised young adults. Given that in our study sample, only very few individuals had severe depressive symptoms, very restricted functional health and/or severe hearing problems, we might have found statistically significant age moderation effects in a clinical sample comprising more individuals with severe hearing loss and with clinically relevant depressive symptoms.

However, the findings supported our assumption that SPA would be more closely related to functional and mental health at older ages, which is in line with previous research [67–70]. Specifically, greater SPA of ongoing development were more closely related to fewer depressive symptoms among chronologically older compared to chronologically younger adults. Similarly, SPA of ongoing development and of physical loss were more closely related to functional health among individuals who were chronologically older. We also found that only among older adults was the association of overall more hearing problems with poorer functional health weaker among those with greater SPA of ongoing development. Finally, and unexpectedly, greater SPA of social loss were associated with less decline in functional health over time among older adults, and this effect was stronger among those with more hearing problems. Again, regression toward the mean might play a role here, as older adults already start from a lower functional health level. Alternatively, those with higher SPA of social loss at baseline might be the ones who try to counteract these loss experiences over time – or they could be the ones who had a higher likelihood of dropping out of the study.

In conclusion, our findings suggest that several SPA are more closely related to mental and functional health at older ages, and that particularly older adults might benefit from a "buffering" effect of positive SPA which seem to reduce to some extent the associations of poorer hearing with poorer functional health.

## Study limitations

This study has several limitations. Specifically, all study variables, including hearing loss, were self-reported so that reporting biases cannot be ruled out. However, in the case of self-reported hearing problems, we have already pointed out that there is a relatively high overlap between self-reported and objective hearing. Moreover, mental health outcomes – such as depressive symptoms, which we investigated in this study – seem to be more reactive to the subjective experience of impaired hearing than to objectively assessed hearing [15–17].

In our study sample of community-dwelling middle-aged and older adults, the prevalence rates of severe hearing problems, severe depressive symptoms or functional health restrictions were very low. Associations of hearing and SPA with mental and functional health might have been stronger in (clinical) samples including, for instance, more individuals with severe hearing loss or clinically relevant depressive symptoms. Also, the associations we identified in the general population may not generalize to specific vulnerable subgroups such as nursing home residents.

Even though we included all individuals who provided at least one valid observation and used Full Information Maximum Likelihood estimation to avoid biased estimates, there might have been selective study attrition. Therefore, we might have underestimated the extent of decline in functional health as well as the extent of increase in depressive symptoms over time.

As previously stated, significant within-person association indicate time-varying associations, but they cannot address causality. Other analytical approaches, including experimental designs, are required in future research to determine whether hearing problems may lead to poorer mental or functional health or vice versa.

Our outcome measures were restricted to mental and functional health. Another important component of health is social health [111]. While some items included in the measurement of depressive symptoms addressed interpersonal aspects (e.g., feelings of loneliness), more research is needed to address whether SPA also moderate the impact of impaired hearing on social embeddedness and social participation.

Finally, in the interest of model parsimony we decided to include SPA as time-invariant predictors and moderators. However, SPA change over time and with advancing age [39]; therefore, future research needs to investigate whether changes in SPA, rather than just initial SPA, are also relevant moderators of associations between sensory losses and mental or physical health. For instance, Sargent-Cox, Anstey [112] found with regard to the role of SPA as a time-invariant vs. time-varying indicator that "a single measurement of SPA in late life may be very informative of future long-term vulnerability to health decline and mortality. Furthermore, a dynamic measure of SPA may be indicative of adaptation to age-related changes" (p. 168).

More research is also needed to identify for whom and via which pathways favorable SPA may counteract the association of more hearing problems with poorer mental and physical health.

Finally, in addition to identifying moderators and mediators of the association of SPA with mental and physical health, future research should address whether other SPA beyond the ones included in this study, such as awareness of age-related changes, or also more implicit measures of views on aging, such as subjective age or (future) time perspectives in general, play such a counteracting role.

## Conclusions

In this study, based on a sample of middle-aged and older adults who were assessed over up to 13 years, we found that more hearing problems were associated with poorer mental and functional health both on the between-person level as well as on the within-person level. Moreover, more positive SPA were related to better health outcomes and they moderated to some extent associations between hearing problems and health, with less negative health outcomes among those individuals with poorer self-reported hearing who had more favorable SPA. Several of the associations of SPA with mental and physical health as well as their moderating effects on associations between hearing problems and health outcomes were additionally moderated by chronological age, with stronger effects among chronologically older adults. Future research should further address this moderating role of SPA and the mechanisms underlying the moderation effect and replication that takes changes in SPA into account is needed. Our findings suggest that prevention and treatment of hearing loss, but also promotion of SPA, may be important elements to promote healthy aging – both mentally and physically – in middle-aged and older adults.

## Supporting information

**S1 File.  Analytical code.**
(DOCX)

## Author contributions

**Conceptualization:** Markus Wettstein, Ann-Kristin Reinhard, Bettina Williger, Susanne Wurm.

**Formal analysis:** Markus Wettstein, Susanne Wurm.

**Methodology:** Markus Wettstein.

**Supervision:** Susanne Wurm.

**Visualization:** Markus Wettstein.

**Writing – original draft:** Markus Wettstein, Ann-Kristin Reinhard, Bettina Williger, Susanne Wurm.

**Writing – review & editing:** Markus Wettstein, Ann-Kristin Reinhard, Bettina Williger, Susanne Wurm.

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
