## [Editor Report · Decision Letter 0]

17 Oct 2024

PONE-D-24-42847Associations of Self-Reported Hearing Problems with Long-Term Trajectories of Mental and Functional Health in Middle-Aged and Older Adults: The Role of Self-Perceptions of AgingPLOS ONE

Dear Dr. Wettstein,

Thank you for submitting your manuscript to PLOS ONE. After careful consideration, we feel that it has merit but does not fully meet PLOS ONE’s publication criteria as it currently stands. Therefore, we invite you to submit a revised version of the manuscript that addresses the points raised during the review process.

Please provide written evidence from an ethical review board that ethics was not required for this survey of human participants before we can proceed to a formal review.

We look forward to receiving your revised manuscript.

Kind regards,

Soham Bandyopadhyay

Academic Editor

PLOS ONE

Journal Requirements:

2. Thank you for stating the following financial disclosure: "The German Ageing Survey (DEAS) was funded under Grant 301-6083-05/003*2 by the German Federal Ministry for Family, Senior Citizens, Women, and Youth."

3. Thank you for stating the following in the Acknowledgments Section of your manuscript: "The German Ageing Survey (DEAS) was funded under Grant 301-6083-05/003*2 by the German Federal Ministry for Family, Senior Citizens, Women, and Youth. The content is the sole responsibility of the authors."

Please remove any funding-related text from the manuscript and let us know how you would like to update your Funding Statement. Currently, your Funding Statement reads as follows: "The German Ageing Survey (DEAS) was funded under Grant 301-6083-05/003*2 by the German Federal Ministry for Family, Senior Citizens, Women, and Youth."

---

## [Decision Letter · Decision Letter 1]

25 Feb 2025

PONE-D-24-42847R1Associations of Self-Reported Hearing Problems with Long-Term Trajectories of Mental and Functional Health in Middle-Aged and Older Adults: The Role of Self-Perceptions of AgingPLOS ONE

Dear Dr. Wettstein,

Thank you for submitting your manuscript to PLOS ONE. After careful consideration, we feel that it has merit but does not fully meet PLOS ONE’s publication criteria as it currently stands. Therefore, we invite you to submit a revised version of the manuscript that addresses the points raised during the review process.

Two expert reviewers and myself evaluated the submitted work. In addition to the detailed comments by the reviewers that primarily require clarifications and consideration of the aspect of social health, I would like you to consider some additional issues, mainly related (but not only restricted) to statistics and data. In short, these refer to the amount of statistical testing and some peculiarities in the figures that drew my attention. Please see below for detailed comments. 

We look forward to receiving your revised manuscript.

Kind regards,

Dimitris Voudouris

Academic Editor

PLOS ONE

Journal Requirements:

The reporting of this study should follow the STROBE checklist (www.strobe-statement.org); for studies with routinely-collected data, please use the RECORD checklist (www.record-statement.org).

**Additional Editor Comments:**

1) The statistical analysis leads to a large amount of statistical tests, which can lead to Type I Errors. At the moment, it does not appear that any corrections are applied, which I would like you to consider. Please also ensure that the conducted statistical tests should address the research hypotheses as these are laid out in the Introduction. Statistical tests for exploratory reasons or for describing data patterns may be removed.  

2) Some statements require clarifications (please see also some comments by Reviewer 1). For instance, the statement "Moreover, people who perceived higher levels of social support were more likely to use hearing aids" can be misinterpreted, as it now reads as a causal relationship. Moreover, in the sentence "Finally, Levy, Slade (55) found a protective effect of positive SPA on a lower dementia risk", how confident can one be that the positive SPA is indeed the mediating factor --and not other factors that are correlated with SPA? Other parts read a bit harder (e.g., last sentence before caption of Fig.2), which I would like you to have one more look at. Please also consider reminding the reader (or explaining more explicitly) what exactly "the buffering role of SPA" mean.

3) Please explain how can one distinguish positive SPA stemming from older adults simply being fitter --somatically/physiologically-- and others not being that fit, and thus have a poorer self-perception.

4) At parts, the term "SPA at baseline" is mentioned. Can you please highlight what exactly this means? In the Discussion, it is mentioned that SPA scores may change throughout the recorded period, which I believe is a limitation that should be considered also in the Abstract --especially since the manuscript examines exactly the role of SPA. 

5) There are numbers and percentages regarding sample size and citizenship. It is unclear what the values 19745 and 1921 refer to: as far as I understand, these do not refer to the data of the current analysis. Can you please clarify? In addition, according to my calculations, the 95.3% of the sum of these two values (19745+1921=21666) is not 19745 but 20647. Instead, the 19745 German nationals are the 91.1% of the sum (21666). Please double-check the values, or correct me if I am wrong.

6) Figure 1: Please edit the figure to explain the x-axis (what are the values 1-4), and please add a legend about the black and red lines. In addition, why do the red lines stop a bit more than halfway between x-axis values 3 and 4, and why do the black lines not stop at x-axis value 4? I expected both lines to span between x=1 and x=4. I also noticed that some individual data are not connected to other points, for instance the datapoint at x=2 in the left panel. Can you please explain why? Generally, I would also recommend adding A and B to the left and right panel, respectively, and referring to those panels when presenting the respective results. This will help the reader follow the main findings without thinking too much about what part of Fig. 1 the sentence refers to. 

7) Last sentence before caption of Figure 1: does this sentence refer to the red or the black line in panel B? And where is the result of the other line --that is supposed to reflect a more systematic relationship?

8) Figure 2: Why do the y-axis values are between 0-10 and those in Figure 1 (left panel) are between 0-40, although both axes show "Depressive Symptoms" of the same sample?

9) Please check the citation style throughout, as it is not always consistent (sometimes they appear in a numerical style, sometimes with author names). 

Reviewers' comments:

Reviewer's Responses to Questions

**Comments to the Author**

1. If the authors have adequately addressed your comments raised in a previous round of review and you feel that this manuscript is now acceptable for publication, you may indicate that here to bypass the “Comments to the Author” section, enter your conflict of interest statement in the “Confidential to Editor” section, and submit your "Accept" recommendation.

Reviewer #1: (No Response)

Reviewer #2: (No Response)

2. Is the manuscript technically sound, and do the data support the conclusions?

Reviewer #1: Yes

Reviewer #2: Yes

3. Has the statistical analysis been performed appropriately and rigorously? 

Reviewer #1: Yes

Reviewer #2: Yes

4. Have the authors made all data underlying the findings in their manuscript fully available?

Reviewer #1: Yes

Reviewer #2: Yes

5. Is the manuscript presented in an intelligible fashion and written in standard English?

Reviewer #1: Yes

Reviewer #2: Yes

6. Review Comments to the Author

Reviewer #1: Thank you for the opportunity to review this insightful and well-conducted study. The authors use data from the German Ageing Survey to test if self-perceptions of aging (SPA; AgeCog scale: ongoing development, social losses, functional losses) moderate the effects of self-reported hearing problems on trajectories of mental health (i.e., depressive symptoms) and functional health (SF36 subscale physical functioning). They found that reported hearing problems were associated with worse health outcomes and that more favourable SPA and chronological age moderated this association both on the interindividual and intra-individual level.

I enjoyed reading this piece and think the paper could be a valuable contribution towards addressing a highly prevalent health concern in midlife and old age. However, I would invite the authors to consider the following minor issues to strengthen their presentation and interpretation of results.

In the revised version of the manuscript, page numbers are lost for page 1 through 15 of the R1 version. Therefore, I refer to PDF page numbers in the following:

1. PDF page1 Abstract, last sentence: to avoid misunderstandings, please be clear if this is meant as a summary of the findings for effects of less negative SPA. If not, I suggest to use SPA “ongoing development” or the like explicitly to refer to the third AgeCog subscale.

2. PDF page 50: “increase” may be a better term than “augment” here, because augment may imply additional mechanisms/aspects

3. PDF page 51: third paragraph: “personality” may be dropped because authors do not cite a study that explicitly looked at personality

4. Fourth paragraph: the fact that SPA are related to moderators does not in itself make them prime candidates for moderating the hearing impairment – health relationship. I wondered if some AgeCog scales (social loss, function loss) could be more specific to the presented moderators of the hearing impairment - health relationship, whereas ongoing development could imply development of social support, function and coping?

5. Heading “The Relevance of Self-Perceptions of Aging”: I would suggest to make this heading more specific by adding “for health outcomes”

6. PDF page 53, last paragraph: I suggest to avoid the term “indirect” here for clarity

7. Last paragraph, second sentence: I found it hard to figure out what authors wanted to imply here. Is the reporting of different quality or validity across different age segments? or are discrepancies between objective hearing impairment and hearing complaints expected to be indicative of different "age expectation" of hearing impairment? Please explain.

8. PDF page 54, “The Present Study”: It is unclear to what extent the social and physical aspects of losses are represented in the cited works on the link between SPA and health and if these would allow for more specific/different hypotheses regarding the (moderating) effects of SPA social loss versus functional loss versus ongoing development. (see also point 4)

9. PDF page 55: put punctuation marks inside quotation marks ." also, add page for direct citation

10. PDF page 57, third paragraph: unclear if data used in this study are generally restricted to 2008-2017 or if only the measure of depressive symptoms at 2020 was not used (similar to the unavailable info on hearing problems and functional health). Please reformulate to avoid misunderstandings

11. Measures section: please check levels of subheadings

12. Mental health: please make clear that the information given in parentheses is is an example item

13. PDF page 59 Statistical Analyses: Please indicate if weights have been used in analyses.

14. PDF page 61 Table 1: please check if these figures are correct and complete. the mean/sd for education appears to have been left out.

15. PDF page 64 last sentence: I wondered if it would be more straightforward to frame this effect as a moderation effect of SPA on the bphearing-time association?

16. PDF page 65 last paragraph: for clarity, authors may want to add "at baseline " to first sentence and then begin with a new sentence

17. PDF page 67 second paragraph: while technically correct, I am not sure if the mean level of several hearing impairment measurements many years apart qualifies as "usual" level of hearing impairment. Authors may use less fraught terms such as “above-average” or the like

18. Replace hypothesis with hypotheses

19. PDF page 68 first sentence: delete opening parenthesis

20. PDF page 69 first paragraph: because authors refer to potential selectivity effects multiple times when discussing their findings, it would be valuable to learn if part of the dropout risk/selectivity has been or could have been addressed/mitigated by using longitudinal weights

21. Authors stress the need to further analyze the role of time-varying SPA. In addition, considering measures that more directly refer to perceptions of age-related change may be another avenue worth mentioning.

22. PDF page 70 second paragraph: drop comma after study sample

Reviewer #2: This meanuscript is very well-written and makes an interesting and novel contribution to the topic of SPA in the context of hearing problems and mental/functional health dimensions.

I only have one major comment, asn some very minor issues.

Major point:

Alongside mental and functional/physical health, the third dimension of social health has already been proposed by the WHO (Triparte Model of health) in 1964 ("Health is a state of complete physical, mental and social well-being...") and has gained more attention recently (e.g., Holt-Lunstadt et al., 2018; Doyle & Link, 2024).

In the field of hearing loss/problems, this should also be included, if possible, as hearing problems have been shown to affect social relationships (or at least be discussed, if not available in the survey). Are there indicators such as the absence of loneliness or presence of connection (e.g. quality and quantity of relationships) in the German Ageing Survey that could be used for social health?

If not, it should at least be discussed as a limitation and an outlook should be given for future study designs (which was generally missing a little in my opinion).

Minor isues:

Table 1:

It was not clear to me, why there are % in the variable names (left column) for the dummy coded variables sex, East vs. West, and the three study entry variables. Please explain in the notes. I would expect an explanation such as: Sex: women = 0, men = 1 (or the other way around), East = …, not %, which is only used in the line containing the means (M) if I understand correctly? The education variable also needs explanation in the note, otherwise, the % cannot be interpreted. Moreover, the levels (1-4) of education need to be explained in the method section. For hearing aid, there are %values missing in the line of the means (M) which should better be named “M / %” or similar. Please also specify in the note how hearing aid was coded (0 = not hearing aid…).

Discussion: Related to my comment on social health, this might be also discussed drawing on Socioemotional Selectivity Theory, hearing loss might be perceived as even more burdensome as the focus is assumed to shift to close social relationships as the time horizon gets limited / people age? Did the authors also consider using subjective age instead of chronological age or future time perspective, if available in the data? Or discuss these issues?

7. PLOS authors have the option to publish the peer review history of their article (what does this mean? ). If published, this will include your full peer review and any attached files.

**Do you want your identity to be public for this peer review?** For information about this choice, including consent withdrawal, please see our Privacy Policy .

Reviewer #1: No

Reviewer #2: **Yes: ** Laura Schmidt

---

## [Author Response · Author response to Decision Letter 2]

4 Apr 2025

PONE-D-24-42847R1

Associations of Self-Reported Hearing Problems with Long-Term Trajectories of Mental and Functional Health in Middle-Aged and Older Adults: The Role of Self-Perceptions of Aging

PLOS ONE

April, 2025

Dear Dr Voudouris,

Thank you very much for your decision on our manuscript. We are very grateful for the opportunity to revise the manuscript and we would like thank you, the additional editor and the reviewers for the feedback. We have tried to address all comments to the best of our abilities, and we believe that your suggestions and feedback have helped us a lot to improve and strengthen our manuscript.

Notes regarding the specific revisions made in response to each comment are included below (indicated by “Response: …”). We hope that you will find the revised manuscript improved and responsive to the queries that had been made. We are looking forward to hearing from you.

Sincerely yours,

On behalf of the authors

Markus Wettstein, PhD

Additional Editor Comments:

Comment: “The statistical analysis leads to a large amount of statistical tests, which can lead to Type I Errors. At the moment, it does not appear that any corrections are applied, which I would like you to consider. Please also ensure that the conducted statistical tests should address the research hypotheses as these are laid out in the Introduction. Statistical tests for exploratory reasons or for describing data patterns may be removed.”

Response: We agree with your point that Type I Error inflation needs to be avoided. Our overall number of statistical models (one with depressive symptoms as outcome, one with functional health as outcome) is only two, but given the quite large sample size, we decided to use a more conservative significance threshold of p less than .01 (see p. 15).

We do not use statistical tests for describing data patterns or for exploratory reasons.

Comment: “Some statements require clarifications (please see also some comments by Reviewer 1). For instance, the statement "Moreover, people who perceived higher levels of social support were more likely to use hearing aids" can be misinterpreted, as it now reads as a causal relationship. Moreover, in the sentence "Finally, Levy, Slade (55) found a protective effect of positive SPA on a lower dementia risk", how confident can one be that the positive SPA is indeed the mediating factor --and not other factors that are correlated with SPA? Other parts read a bit harder (e.g., last sentence before caption of Fig.2), which I would like you to have one more look at. Please also consider reminding the reader (or explaining more explicitly) what exactly "the buffering role of SPA" mean.”

Response:

Thank you for raising this point. We agree that firm causal conclusions cannot be drawn from longitudinal observational studies. We have removed the statement on social support and hearing aids, as it was not crucial for our research question. We have also changed the summary of the findings by Levy et al. accordingly into “Finally, Levy et al. (50) found that more positive SPA at baseline were associated with a lower risk of developing dementia over an observation period of four years, controlling for a broad set of relevant covariates such as depressive symptoms, cardiovascular diseases or diabetes.” (p. 6).

We have changed the last sentence before the Fig. 2 caption and split it into two sentences to facilitate the interpretation (p. 20).

On p. 24 we explain in more detail what “the buffering role of SPA” indicates (“The role of SPA as `buffer` in the context of health risk factors, with positive SPA being associated with a reduced detrimental impact of risk factors on health outcomes”).

Comment: “3) Please explain how can one distinguish positive SPA stemming from older adults simply being fitter --somatically/physiologically-- and others not being that fit, and thus have a poorer self-perception.”

Response: We agree that the associations between SPA and health are very likely bidirectional. However, the role of SPA as predictor of subsequent health seems to be stronger than the role of health as a predictor of SPA (see for instance Levy et al., 2002, Sargent-Cox et al., 2012, Tovel et al., 2019, Wurm et al., 2007). Moreover, when investigating the role of SPA, studies typically control for health indicators (see for instance in the Levy et al. study listed on p. 6), and we did the same in this study by controlling for chronic diseases.

Comment: “At parts, the term "SPA at baseline" is mentioned. Can you please highlight what exactly this means? In the Discussion, it is mentioned that SPA scores may change throughout the recorded period, which I believe is a limitation that should be considered also in the Abstract --especially since the manuscript examines exactly the role of SPA.”

Response: Thank you, we agree that considering SPA as time-invariant, which we did to avoid an overcomplex model with too many time-varying predictors (in addition to time-varying hearing problems), is a study limitation, which we now explicitly point out on p. 27. We have now also made more clear, both in the abstract and in the main text (p. 2, p. 12), that the assessment in 2008 corresponds to the “baseline”.

Comment: “There are numbers and percentages regarding sample size and citizenship. It is unclear what the values 19745 and 1921 refer to: as far as I understand, these do not refer to the data of the current analysis. Can you please clarify? In addition, according to my calculations, the 95.3% of the sum of these two values (19745+1921=21666) is not 19745 but 20647. Instead, the 19745 German nationals are the 91.1% of the sum (21666). Please double-check the values, or correct me if I am wrong.”

Response: To avoid confusion, we now report the percentages only for our study sample and no longer for all individuals who ever participated (at least once) in the German Ageing Survey (p. 12).

Regarding your second question, please note that having a German citizenship and having a migration background is not mutually exclusive, so the two values are not supposed to be summed up.

Comment: “6) Figure 1: Please edit the figure to explain the x-axis (what are the values 1-4), and please add a legend about the black and red lines. In addition, why do the red lines stop a bit more than halfway between x-axis values 3 and 4, and why do the black lines not stop at x-axis value 4? I expected both lines to span between x=1 and x=4. I also noticed that some individual data are not connected to other points, for instance the datapoint at x=2 in the left panel. Can you please explain why? Generally, I would also recommend adding A and B to the left and right panel, respectively, and referring to those panels when presenting the respective results. This will help the reader follow the main findings without thinking too much about what part of Fig. 1 the sentence refers to.

Response: Please see our revised figure caption and figure legend where we now explain the x-axis scores and the black and red lines (p. 18). We have changed the length of the lines so that they cover the entire x-axis span. Single green points which are not connected to other points represent individuals who only took part once in the survey and thus provided no longitudinal information. Thank you for your suggestion of adding “A” and “B” to the figures for a better discernibility, we have added these letters accordingly, also for the other figures.

Comment: “Last sentence before caption of Figure 1: does this sentence refer to the red or the black line in panel B? And where is the result of the other line --that is supposed to reflect a more systematic relationship?”

Response: This last sentence refers to a (nonsignificant) effect that is not illustrated in Figure 1. The sentences before this sentence refer to the black and red lines in Figure 1A, and both lines represent significant associations. We have now made these effects and their illustration in Figure 1 a more explicit (p. 18).

Comment: “Figure 2: Why do the y-axis values are between 0-10 and those in Figure 1 (left panel) are between 0-40, although both axes show "Depressive Symptoms" of the same sample?”

Response: Thank you for your thorough observation. It is true that depressive symptoms ranged from 0-42 (see also Table 1). For Figure 1A, where depressive symptoms of each individual are displayed, we decided to show the full range of depressive symptoms. In Figure 2, where mean depressive symptoms are shown by age group and by SPA of ongoing development, we decided not to show the entire depressive symptoms range, as this might make it harder to detect the relevant differences according to age and SPA of ongoing development (see also our added comment in the Figure legend on p. 20). However, if preferred, we can widen the axis scale range of depressive symptoms for Figure 2.

Comment: “Please check the citation style throughout, as it is not always consistent (sometimes they appear in a numerical style, sometimes with author names).”

Response: We apologize for these inconsistencies, which we have corrected throughout the manuscript.

Reviewer #1:

Comment: „Thank you for the opportunity to review this insightful and well-conducted study. The authors use data from the German Ageing Survey to test if self-perceptions of aging (SPA; AgeCog scale: ongoing development, social losses, functional losses) moderate the effects of self-reported hearing problems on trajectories of mental health (i.e., depressive symptoms) and functional health (SF36 subscale physical functioning). They found that reported hearing problems were associated with worse health outcomes and that more favourable SPA and chronological age moderated this association both on the interindividual and intra-individual level.

I enjoyed reading this piece and think the paper could be a valuable contribution towards addressing a highly prevalent health concern in midlife and old age. However, I would invite the authors to consider the following minor issues to strengthen their presentation and interpretation of results.”

Response: Thank you very much for your positive feedback and your very helpful suggestions to improve this manuscript.

Comment: “In the revised version of the manuscript, page numbers are lost for page 1 through 15 of the R1 version. “

Response: We apologize for this mistake, we have added page numbers accordingly.

Comment: “PDF page1 Abstract, last sentence: to avoid misunderstandings, please be clear if this is meant as a summary of the findings for effects of less negative SPA. If not, I suggest to use SPA “ongoing development” or the like explicitly to refer to the third AgeCog subscale.”

Response: We agree that this statement was not sufficiently precise; we have changed the sentence accordingly (“Our findings suggest that positive SPA related to fewer physical losses and to more ongoing development may buffer the negative impact of hearing problems on functional health.”).

Comment: “2. PDF page 50: “increase” may be a better term than “augment” here, because augment may imply additional mechanisms/aspects.”

Response: We have changed the wording accordingly (p. 4).

Comment: ”3. PDF page 51: third paragraph: “personality” may be dropped because authors do not cite a study that explicitly looked at personality.”

Response: Thank you for this suggestion, we now do not refer to personality anymore in that paragraph.

Comment: ”4. Fourth paragraph: the fact that SPA are related to moderators does not in itself make them prime candidates for moderating the hearing impairment – health relationship. I wondered if some AgeCog scales (social loss, function loss) could be more specific to the presented moderators of the hearing impairment - health relationship, whereas ongoing development could imply development of social support, function and coping?”

Response: Our idea, which we now made more explicit on p. 5, is that SPA might operate as moderators via other mechanisms/mediators (e.g. coping strategies), even though we could not empirically test mediators of the SPA moderation. Thank you for raising the point that different AgeCog scales reveal different associations with factors such as social support; we now refer to this important differentiation on p. 5 and pp. 10-11.

Comment: “Heading “The Relevance of Self-Perceptions of Aging”: I would suggest to make this heading more specific by adding “for health outcomes”.”

Response: Thank you for your suggestion, we have changed the heading accordingly (p.5).

Comment: “6. PDF page 53, last paragraph: I suggest to avoid the term “indirect” here for clarity.”

Response: We have removed the term “indirect” and rewritten the sentence accordingly (p. 7).

Comment: “7. Last paragraph, second sentence: I found it hard to figure out what authors wanted to imply here. Is the reporting of different quality or validity across different age segments? or are discrepancies between objective hearing impairment and hearing complaints expected to be indicative of different "age expectation" of hearing impairment? Please explain.”

Response: We refer in this sentence to the discrepancy between self-reported and objective hearing loss, which is, as reported by Bainbridge and Wallhagen (2014), different in middle-aged and older adults. We agree that our prior description might have been misleading and not clear enough. Therefore, we have changed the sentence accordingly (“Among middle-aged adults, self-reported hearing loss is more frequent than audiometrically assessed hearing loss, whereas among older adults, the proportion of individuals affected by audiometrically assessed hearing loss is greater than the proportion of individuals with self-reported hearing loss (3)”; pp. 7-8). We have also added a potential (but speculative) explanation for these age differences (pp. 7-8).

Comment: “8. PDF page 54, “The Present Study”: It is unclear to what extent the social and physical aspects of losses are represented in the cited works on the link between SPA and health and if these would allow for more specific/different hypotheses regarding the (moderating) effects of SPA social loss versus functional loss versus ongoing development. (see also point 4).”

Response: Thank you for raising this point. Given the scarcity of prior findings, particularly of studies with a differentiation of gain- vs. loss-oriented SPA, and as we ran the analyses already, we decided not to add specific hypotheses in a post-hoc manner. However, we added the possibility of SPA-domain-specific effects as an exploratory research question on pp. 10-11. In the introduction section, whenever one of the AgeCog scales is part of described previous findings, we have added the information which specific AgeCog scale was investigated (p. 6-p. 7). However, most of the prior studies did not include multidimensional SPA measures and rather relied on unidimensional measures such as the Attitude Toward Own Aging Scale.

Comment: “9. PDF page 55: put punctuation marks inside quotation marks ." also, add page for direct citation.”

Response: Thank you, we have changed this accordingly (p. 9; p. 27).

Comment: “10. PDF page 57, third paragraph: unclear if data used in this study are generally restricted to 2008-2017 or if only the measure of depressive symptoms at 2020 was not used (similar to the unavailable info on hearing problems and functional health). Please reformulate to avoid misunderstandings.”

Response: We now explain in more detail that the summer measurement occasion 2020 was an assessment with focus on pandemic-related appraisals and experiences (p. 12); as several of our study variables were not assessed in summer 2020, we did not include this measurement occasion, so the measurements occasions included were 2008, 2011, 2014, 2017 and winter 2020/2021.

Comment: “11. Measures section: please check levels of subheadings.”

Response: Thank you very much for your thorough reading. We have adjusted the subheading levels accordingly.

Comment: “12. Mental health: please make clear that the information given in parentheses is an example item.”

Response: Thank y

---

## [Decision Letter · Decision Letter 2]

8 Jul 2025

Associations of Self-Reported Hearing Problems with Long-Term Trajectories of Mental and Functional Health in Middle-Aged and Older Adults: The Role of Self-Perceptions of Aging

PONE-D-24-42847R2

Dear Dr. Wettstein,

We’re pleased to inform you that your manuscript has been judged scientifically suitable for publication and will be formally accepted for publication once it meets all outstanding technical requirements. 

Kind regards,

Dimitris Voudouris

Academic Editor

PLOS ONE

Additional Editor Comments (optional):

Reviewers' comments:

Reviewer's Responses to Questions

**Comments to the Author**

1. If the authors have adequately addressed your comments raised in a previous round of review and you feel that this manuscript is now acceptable for publication, you may indicate that here to bypass the “Comments to the Author” section, enter your conflict of interest statement in the “Confidential to Editor” section, and submit your "Accept" recommendation.

Reviewer #1: All comments have been addressed

2. Is the manuscript technically sound, and do the data support the conclusions?

Reviewer #1: Yes

3. Has the statistical analysis been performed appropriately and rigorously? 

Reviewer #1: Yes

4. Have the authors made all data underlying the findings in their manuscript fully available?

Reviewer #1: Yes

5. Is the manuscript presented in an intelligible fashion and written in standard English?

Reviewer #1: Yes

6. Review Comments to the Author

Reviewer #1: Thank you for the opportunity to follow the progress of this manuscript submitted to PlosOne. The authors’ response to the concerns raised by reviewers and the editor was comprehensive, thoughtful, and effective. Reading the revised manuscript, the revisions clarified and strengthened the presentation and interpretation of results. The authors’ decision not to broaden the scope of their current analysis but to address ideas for both more specific associations with respect to facets of SPA and social health in the discussion seems reasonable.

However, the following editorial changes might be necessary:

1. Page 3 paragraph 3: abbreviation HL in the second set of parentheses has not been explained before. Rather than explaining this within the first set of parentheses, I suggest spelling out hearing level.

2. Page 6 second paragraph, last sentence: delete the period within the sentence

3. Page 7 second paragraph: APA7 guidelines does not suggest capitalization of theories in text

4. Page 10 second paragraph: The last sentence detailing inclusion criteria for waves not used in this study may be dropped to save words.

5. Table 1: in line (14) please add % before hearing aid for consistency. In line (15), please indicate range of chronic diseases for consistency

6. Page 20 first paragraph: “With respect to” instead of “in relation to”?

7. Page 20 second paragraph: The first sentence could be deleted, as it currently also references effects of hearing problems for which no significant interactions have been found.

8. Page 20 last paragraph: use of hearing aids was not a significant (at p<.01) predictor of depressive symptoms. Similarly, the Study entry 2014*time interaction is no longer statistically significant. Please drop.

9. Page 22 third paragraph: The Study entry 2008*time interaction is no longer statistically significant. Please drop.

10. Page 23 first paragraph: add missing closing parenthesis

7. PLOS authors have the option to publish the peer review history of their article (what does this mean? ). If published, this will include your full peer review and any attached files.

**Do you want your identity to be public for this peer review?** For information about this choice, including consent withdrawal, please see our Privacy Policy .

Reviewer #1: No

---

## [Editor Report · Acceptance letter]

PONE-D-24-42847R2

PLOS ONE

Dear Dr. Wettstein,

I'm pleased to inform you that your manuscript has been deemed suitable for publication in PLOS ONE. Congratulations! Your manuscript is now being handed over to our production team.

Kind regards,

on behalf of

Dr. Dimitris Voudouris

Academic Editor

PLOS ONE